# Cardiometabolic multimorbidity and associated patterns of healthcare utilization and quality of life: Results from the Study on Global AGEing and Adult Health (SAGE) Wave 2 in Ghana

Peter Otieno[1,2,3]*, Gershim Asiki[1,4], Calistus Wilunda[1], Welcome Wami[3,5], Charles Agyemang[2]

1 African Population and Health Research Center, Nairobi, Kenya, 2 Department of Public & Occupational Health, Amsterdam Public Health Research Institute, Amsterdam UMC, University of Amsterdam, Amsterdam, The Netherlands, 3 Amsterdam Institute for Global Health and Development, Amsterdam, The Netherlands, 4 Department of Women's and Children's Health, Karolinska Institutet, Stockholm, Sweden, 5 Department of Global Health, University of Amsterdam, Amsterdam, The Netherlands

* pootienoh@gmail.com

**Data Availability Statement:** Data from SAGE Ghana Wave 2 was used for this study. The

## Abstract

Understanding the patterns of multimorbidity, defined as the co-occurrence of more than one chronic condition, is important for planning health system capacity and response. This study assessed the association of different cardiometabolic multimorbidity combinations with healthcare utilization and quality of life (QoL). Data were from the World Health Organization (WHO) study on global AGEing and adult health Wave 2 (2015) conducted in Ghana. We analysed the clustering of cardiometabolic diseases including angina, stroke, type 2 diabetes, and hypertension with unrelated conditions such as asthma, chronic lung disease, arthritis, cataract and depression. The clusters of adults with cardiometabolic multimorbidity were identified using latent class analysis and agglomerative hierarchical clustering algorithms. We used negative binomial regression to determine the association of multimorbidity combinations with outpatient visits. The association of multimorbidity clusters with hospitalization and QoL were assessed using multivariable logistic and linear regressions. Data from 3,128 adults aged over 50 years were analysed. We identified four distinct classes of multimorbidity: relatively "healthy class" with no multimorbidity (47.9%): abdominal obesity only (40.7%): cardiometabolic and arthritis class comprising participants with hypertension, type 2 diabetes, stroke, abdominal and general obesity, arthritis and cataract (5.7%); and cardiopulmonary and depression class including participants with angina, chronic lung disease, asthma, and depression (5.7%). Relative to the class with no multimorbidity, the cardiopulmonary and depression class was associated with a higher frequency of outpatient visits [β = 0.3; 95% CI 0.1 to 0.6] and higher odds of hospitalization [aOR = 1.9; 95% CI 1.0 to 3.7]. However, cardiometabolic and arthritis class was associated with a higher frequency of outpatient visits [β = 0.8; 95% CI 0.3 to 1.2] and not hospitalization [aOR = 1.1; 95% CI 0.5 to 2.9]. The mean QoL scores was lowest among participants in the cardiopulmonary and

necessary permission was obtained from the World Health Organization to use the data. All files were obtained from the World Health Organization Study on global AGEing and adult health (WHO-SAGE). Details on data can be found at http://www.who.int/healthinfo/sage/cohorts/en/. The authors confirm that they had no special access privileges to the data. Interested researchers will have to submit a licensed data request to WHO. Upon approval, the researchers will be granted access to licensed data.

**Funding:** Financial support was provided by the US National Institute on Aging through Interagency Agreements (OGHA 04034785; YA1323-08-CN-0020;Y1-AG-1005-01) with the World Health Organization and a Research Project Grant (R01 AG034479- 64401A1). The funders had no role in study design, data collection and analysis, decision to publish, or preparation of the manuscript.

**Competing interests:** The authors have declared that no competing interests exist.

depression class [β = -4.8; 95% CI -7.3 to -2.3] followed by the cardiometabolic and arthritis class [β = -3.9; 95% CI -6.4 to -1.4]. Our findings show that cardiometabolic multimorbidity among older persons in Ghana cluster together in distinct patterns that differ in healthcare utilization. This evidence may be used in healthcare planning to optimize treatment and care.

## Introduction

Sub-Saharan Africa is undergoing more rapid ageing than high-income countries [1]. This poses potential critical challenges for older persons, central among them is the burden of chronic diseases [2]. People living with chronic conditions often have multiple rather than a single condition, commonly referred to as multimorbidity [3]. In Ghana, three in every five older persons aged above 50 years live with multimorbidity [4]. Cardiometabolic diseases such as hypertension and type 2 diabetes account for the highest burden of multimorbidity in Ghana [5]. Importantly, cardiometabolic diseases often coexist with other chronic diseases with unrelated pathophysiology such as mental illnesses, chronic lung diseases and musculo-skeletal disorders [6–8]. This phenomenon is referred to as discordant multimorbidity [9].

The management of multimorbidity is complex and demanding for healthcare systems in Ghana [10,11]. This is because the current chronic disease management guidelines were developed when having a single chronic disease was common and focused on a single disease [12,13]. The recent World Health Organization [WHO] guidelines on multimorbidity question this single-disease management approach and highlight the need for accounting for all multimorbidity when informing the patient about available treatment options [14]. However, studies conducted in Ghana show that people living with multimorbidity face several challenges such as fragmented appointments, difficulties with access to information, and a lack of coherence or coordination of care [15,16]. Furthermore, therapeutic interventions for multimorbidity are a major challenge due to polypharmacy and poor medication adherence [17]. Integrated management of multimorbidity and a shift of the treatment goals towards medical care that is less disruptive may partly lower the treatment burden [18].

Previous studies show a positive association between the number of co-existing chronic conditions and frequency of outpatient visits, longer hospital stays, and poor health-related quality of life [4,19–22]. However, the multimorbidity counts or indices used in the vast majority of existing studies do not provide adequate information on specific disease clusters to guide integrated care interventions [23,24]. Although the use of disease count is important in establishing the prevalence of multimorbidity, clusters of conditions that tend to co-occur non-randomly is more useful for clinical practice and health policy. Thus, a deeper insight into the multimorbidity burden on healthcare utilization that goes beyond counting the number of coexisting chronic conditions is needed [25]. Understanding multimorbidity clusters and healthcare utilization patterns is important for planning health system capacity and response to optimise healthcare resources and accommodate patient needs.

The aim of this study was to identify classes of adults with cardiometabolic multimorbidity and determine the association of different multimorbidity combinations with healthcare utilization and quality of life (QoL).

## Methods

### Study design

The data for this study are from the WHO Study on Global AGEing and Adult Health (SAGE) Wave 2 survey conducted in Ghana in 2015 [26]. The WHO SAGE aims to provide reliable

evidence on the health and well-being of older persons aged over 50 years in low and middle-income countries [27]. The study design is provided elsewhere [28]. In brief, a stratified multi-stage cluster sampling method was used to collect data from a nationally representative sample of adults aged 50 years and older. Detailed descriptions of sampling methods and data collection procedures have been previously published [28–30].

The original study sample comprised 3,575 older persons aged over 50 years. Participants were included in the current analysis if they had valid data on the key variables: chronic diseases such as angina pectoris, stroke, type 2 diabetes, hypertension, obesity, arthritis, asthma, chronic lung disease, depression, and cataracts and sociodemographic characteristics comprising sex, age, and employment. Participants (n = 447) for which data on key variables were not captured or judged as invalid were excluded. Since the causes of missing information were not ascertained, we did not apply missing data techniques to avoid further uncertainty in the imputation models. Thus, the final analysis included 3,128 participants.

## Data collection

Data used in the current study were collected using interviewer-administered structured questionnaires [31]. Detailed information on the study tools has been published [32]. Data were collected on socio-demographic characteristics, chronic conditions, healthcare utilization and QoL. The chronic conditions comprised cardiometabolic diseases such as angina pectoris, stroke, diabetes, obesity and hypertension, and unrelated conditions such as arthritis, asthma, chronic lung disease, depression, and cataracts.

## Measurement and definition of variables

**Outcome variable.** The outcome variables were frequency of outpatient visits, hospitalization and QoL. The frequency of outpatient visits was measured as the number of times a participant had an outpatient visit in the preceding 12 months. Hospitalization was measured as any overnight stays in the hospital that lasted for at least one night in the past 12 months. An 8-item World Health Organization Quality of Life (WHOQOL) instrument was used to assess the QoL score [33]. The WHOQOL comprises two questions across each of the four main life domains: physical, psychological, social, and environmental [33]. Using a five-point Likert scale, ranging from very satisfied to very dissatisfied, the respondents rated their satisfaction with life domains such as health, ability to perform daily activities and meet basic needs, relationships, and environment. The composite score of QoL is the sum of the 8 items from the four domains expressed as a percentage.

**Explanatory variables.** The main explanatory variable was cardiometabolic multimorbidity defined as the coexistence of at least two cardiometabolic diseases including obesity, angina, stroke, type 2 diabetes, hypertension or a discordant multimorbidity comprising at least one cardiometabolic disease and an unrelated chronic disease such as asthma, chronic lung disease, arthritis, cataract and depression. The multimorbidity clusters were named based on their unique dominant chronic diseases.

Self-reported history of diagnosis by a healthcare professional was extracted for cardiometabolic diseases comprising angina, stroke, type 2 diabetes, hypertension and other conditions such as arthritis, asthma, chronic lung disease, depression, and cataract. The WHO symptomatology algorithms [34–36] were used to screen for angina pectoris, arthritis, asthma, chronic lung disease, and depression. S1 Table shows the details of the symptomatology algorithms. Physical measurements comprised screening for blood pressure (BP) and anthropometrics. Hypertension was defined as systolic BP ≥ 140 mmHg and/or diastolic BP ≥ 90 mmHg or previous diagnosis of hypertension by a professional health care provider and/or being on

hypertensive therapy [37]. Abdominal obesity was defined using WHO guidelines as a waist circumference ≥ 94 cm for men, or ≥ 80 cm for women [38]. General obesity was defined as a body mass index ≥30.0 kg/m² [39].

Other explanatory variables comprised sociodemographic and health characteristics such as sex, age, education, employment, health insurance coverage, primary source of care (private, public, faith-based/charity hospital) and place of residence (urban or rural).

### Data analysis

Descriptive statistics comprising frequencies, means, medians, standard deviations, interquartile range, and Pearson's chi-squared tests were used to summarize the characteristics of the study participants while accounting for sampling weights.

**Latent class analysis.** Latent Class Analysis (LCA) was used to place participants in a number (K) of clinically meaningful classes of cardiometabolic multimorbidity. The number of multimorbidity classes was defined a priori using the adjusted Bayesian information criterion (BIC), a model selection method that balances fit with parsimony [40]. Five plausible LCA models were delineated, characterized by increasing numbers of chronic disease classes from one to five (S2 Table). The model with the lowest value of the BIC index was selected as the best-fitting model considering interpretability and clinical judgment [40,41]. Posterior probabilities were used to determine the likelihood of class membership. Finally, the participants were grouped into the multimorbidity classes with the highest-class probability [38].

**Hierarchical cluster analysis.** We identified clinically meaningful clusters of multimorbidity using agglomerative hierarchical clustering algorithms [39]. Data used in our analysis are a collection of binary objects arranged in an n×p matrix. The rows represent the (n = 3, 128) study participants and the columns represent the (p = 11) chronic diseases including abdominal obesity, hypertension, general obesity, arthritis, asthma, cataract, type 2 diabetes, angina, chronic lung disease, depression and stroke. The classical approach to the analysis of multimorbidity clusters comprises the grouping of "n" study participants into a set of clusters using the proximity index among the study respondents. This yields an "n×n" proximity matrix that reflects the degree of closeness among the study participants and describes the patterns of disease clusters. However, in the current study, we analysed the multimorbidity patterns by clustering the outcome variables i.e. multimorbidity rather than the observations. This approach is more robust than the former since it reduces the transposed "p×n" data matrix to a much smaller "p×p" proximity matrix among the chronic disease outcomes compared to a potentially large "n×n" proximity matrix [42]. First, individual chronic diseases were grouped in a single cluster. Second, the individual disease clusters were gradually merged with the most closely related clusters until a single cluster with all the elements was obtained. To accommodate the spread of the cluster, we used the average linkage method [43]. Finally, we assessed the number of clusters using a dendrogram and Jaccard similarity coefficient [39].

**Regression analysis.** We used negative binomial regression to determine the association of multimorbidity combinations with outpatient visits. Negative binomial regression has inbuilt parameters that account for the overdispersion problem of modelling healthcare utilization frequency [44]. The association of multimorbidity combinations with hospitalization and QoL were assessed using multivariable logistic and linear regressions. Bivariable negative binomial regression, logistic and linear regression with the frequency of outpatient visits, hospitalization, and QoL as the outcome variables, were first fitted for each of the multimorbidity classes followed by a multivariable model adjusting for socio-demographic characteristics namely age, sex, education, employment status, health insurance coverage, and place of residence. Because of the clustered design of the sample, robust variance estimates (Huber-White

sandwich estimator) were used for the correction of standard errors to adjust for the correlation among responses within the same household [45]. The strength of association was interpreted using the adjusted odds ratios (aOR) and 95% confidence intervals (CI) from logistics regression and beta (β) coefficients from negative binomial and linear regressions [46,47]. P values of <0.05 were considered statistically significant.

We assessed the goodness of fit of the bivariate and multivariable models using the likelihood ratio test [48].

All statistical analyses were carried out using Stata 17.0 (StataCorp LP, Texas, USA) and accounted for the complex sampling design used in the WHO SAGE survey.

## Ethics approval and consent to participate

All methods were carried out in accordance with the relevant guidelines and regulations. This study was approved by the World Health Organization's Ethical Review Board (reference number RPC149) and the Ethical and Protocol Review Committee, College of Health Sciences, University of Ghana, Accra, Ghana. The respondents went through an informed consent process and their participation was voluntary and anonymous. Written consent was provided before participation.

## Results

### Characteristics of participants

The sociodemographic and health characteristics of the study participants are presented in Table 1. In total, 3,128 participants were included in the analysis. In general, most of the participants were women, aged between 50–59 years [51.2%], had no formal education [41.5%], self-employed [69.7%], lived in rural areas [52.1%], and sought care from public facilities [41.4%]. Only a quarter of the participants had health insurance coverage. The most prevalent chronic diseases were abdominal obesity [47.0%] and hypertension [37.1%]. The prevalence of abdominal and general obesity, arthritis angina and depression were significantly higher in females than males.

### Findings of Latent Class Analysis

The multimorbidity classes are shown in Fig 1. We compared LCA models with 1 to 5 classes (online S2 Table). The four-class model had the lowest BIC index and thus was selected as the best-fit model. Class one comprised relatively "healthy participants" with no multimorbidity [47.9%]. Class two included participants with a high probability of abdominal obesity only [40.7%]. Class three comprised participants with high probabilities of hypertension, diabetes, stroke, abdominal and general obesity, arthritis and cataract [5.7%]. Class four (cardiopulmonary diseases and depression) comprised participants with high probabilities of angina, chronic lung disease, asthma and depression [5.7%].

### Hierarchical cluster analysis findings

As a supplementary analysis, we used hierarchical cluster analysis with agglomerative algorithms to compute the multimorbidity patterns. Fig 2 shows a dendrogram with a hierarchical tree plot of the multimorbidity clusters. The dendrogram shows a graphical representation of the agglomeration schedules at which multimorbidity clusters are combined. In general, our results were consistent with those obtained using LCA. The hierarchical clustering algorithms revealed distinct groupings of multimorbidity in the study sample. Based on the proximity coefficients, the first cluster comprised angina, chronic lung disease, asthma, and depression

**Table 1. Sociodemographic and health characteristics of the study participants.**

| | Characteristics (%) | Both sexes | Males | Females | P value |
|---|---|---|---|---|---|
| - | N | 3,128 | 1,306 | 1,822 | |
| Age [Years] | | | | | 0.063 |
| | 50–59 | 51.2 | 52.5 | 50.0 | |
| | 60–69 | 27.5 | 28.3 | 26.7 | |
| | 70+ | 21.3 | 19.2 | 23.3 | |
| Education | | | | | <0.001 |
| | No formal education | 41.5 | 29.8 | 52.2 | |
| | Primary | 28.3 | 28.7 | 27.8 | |
| | Secondary | 26.5 | 35.9 | 17.9 | |
| | Tertiary | 3.7 | 5.6 | 2.1 | |
| Employment | | | | | <0.001 |
| | Public | 7.8 | 11.7 | 4.3 | |
| | Private | 4.4 | 6.7 | 2.2 | |
| | Self-employed | 69.7 | 64.8 | 74.2 | |
| | Informal employment | 16.2 | 15.2 | 17.2 | |
| | Unemployed | 1.9 | 1.7 | 2.0 | |
| Place of residence | | | | | 0.785 |
| | Urban | 47.9 | 47.5 | 48.2 | |
| | Rural | 52.1 | 52.5 | 51.8 | |
| Primary source of care | | | | | 0.013 |
| | Private facility | 9.3 | 8.8 | 9.6 | |
| | Public facility | 41.4 | 37.4 | 45 | |
| | Faith-based/charity hospital | 4.0 | 4.3 | 3.7 | |
| | [†] Others | 3.6 | 3.6 | 3.6 | |
| | Never sought care | 41.8 | 45.9 | 38.1 | |
| Health insurance coverage | | | | | |
| | Yes | 25.4 | 21.6 | 28.9 | <0.001 |
| Chronic diseases | | | | | |
| | Abdominal obesity | 47.0 | 16.5 | 74.8 | <0.001 |
| | Hypertension | 37.1 | 36.7 | 37.4 | 0.761 |
| | General obesity | 13.4 | 5.9 | 20.2 | <0.001 |
| | Arthritis | 20.4 | 17.3 | 23.2 | 0.005 |
| | Asthma | 8.3 | 8.0 | 8.6 | 0.634 |
| | Cataract | 7.2 | 6.4 | 7.9 | 0.201 |
| | Diabetes | 2.6 | 2.5 | 2.7 | 0.798 |
| | Angina | 8.4 | 5.0 | 11.4 | <0.001 |
| | Chronic lung disease | 4.5 | 4.1 | 4.8 | 0.343 |
| | Depression | 4.5 | 3.1 | 5.7 | <0.001 |
| | Stroke | 1.2 | 1.1 | 1.3 | 0.502 |

Cells are weighted percentages unless otherwise specified.

[†] Other sources of primary care comprise local pharmacies and traditional healers.

(cardiopulmonary and depression class). The second cluster comprised participants with hypertension, abdominal and general obesity, arthritis, cataract stroke and diabetes (cardiometabolic diseases, arthritis and cataract class).

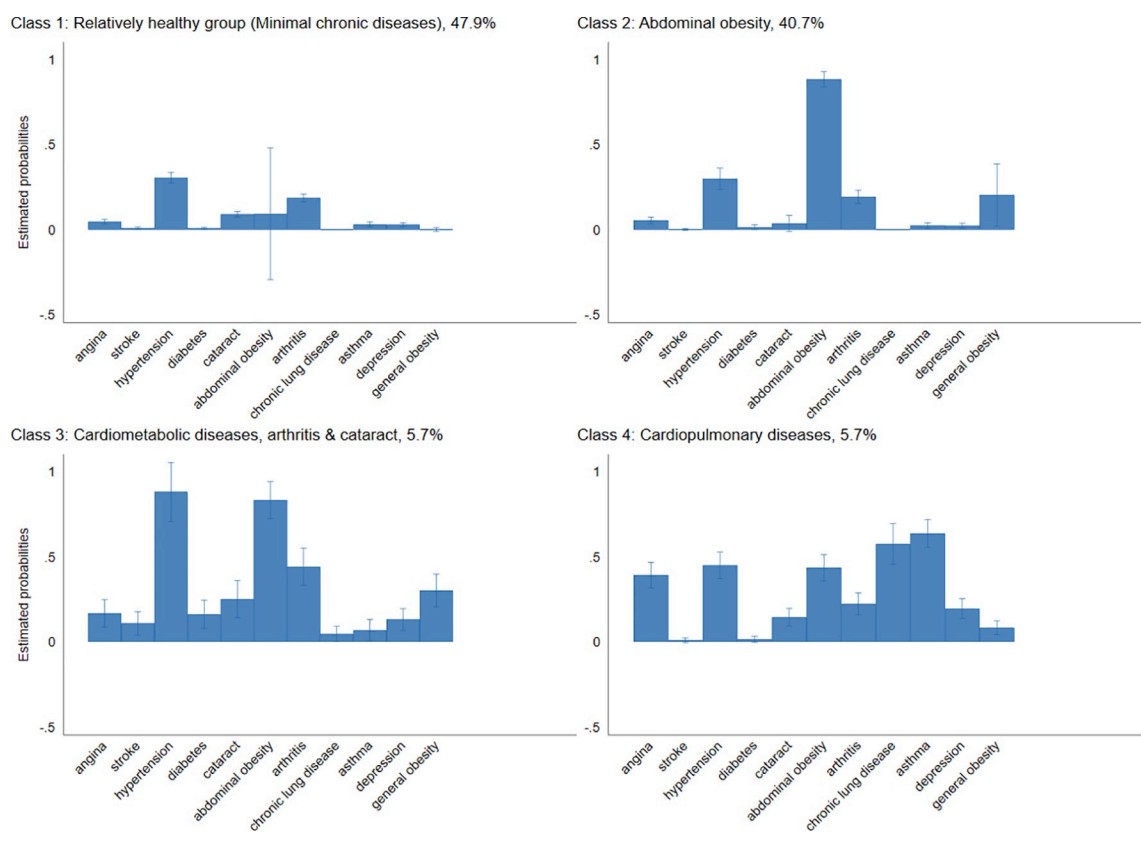

**Fig 1.**

## Sociodemographic distribution of multimorbidity patterns

The sociodemographic distribution of multimorbidity classes is presented in Table 2. The majority of the participants with abdominal obesity were aged between 50 and 59 years. However, most participants with cardiometabolic and cardiopulmonary multimorbidity were older (aged 60–69 years and 70 years and above). Most participants with abdominal obesity and cardiometabolic multimorbidity resided in urban settings while a majority of those with cardiopulmonary multimorbidity resided in rural settings. In general, most of the participants with abdominal obesity and those with cardiometabolic and cardiopulmonary multimorbidity were females, self-employed, had no formal education nor insurance coverage, and sought care from public facilities.

## Frequency of healthcare utilization and quality of life

The patterns of healthcare utilization and QoL is presented in Table 3. In general, the participants who visited outpatient clinics frequently and those hospitalized at least once in the previous 12 months were mostly older, women, lived in urban settings, sought primary care from faith-based or charity organizations, and had cardiometabolic and cardiopulmonary multimorbidity. Other participants who visited outpatient clinics frequently mostly comprised those with tertiary-level of education and health insurance coverage and employed in public or informal settings. The QoL score was lowest among older participants, females, unemployed, those with no formal education nor health insurance coverage, living in urban settings, seeking care from faith-based or charity organizations, and participants with cardiometabolic and cardiopulmonary multimorbidity.

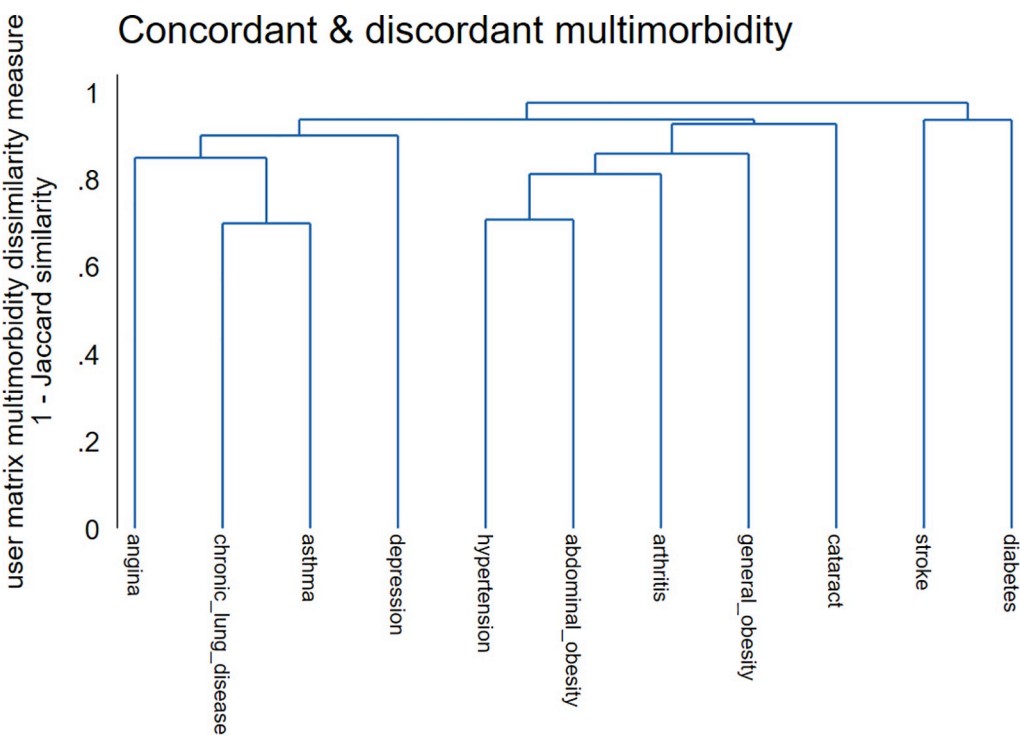

**Fig 2.**

## Cardiometabolic multimorbidity classes and associated healthcare utilization patterns and QoL in Ghana

Fig 3 shows the association of different multimorbidity combinations with healthcare utilization and QoL. Relative to the class with no multimorbidity, the cardiopulmonary and depression class was associated with a higher frequency of outpatient visits [β = 0.3; 95% CI 0.1 to 0.6] and higher odds of hospitalization [aOR = 1.9; 95% CI 1.0 to 3.7]. However, cardiometabolic and arthritis class was associated with a higher frequency of outpatient visits [β = 0.8; 95% CI 0.3 to 1.2] and not hospitalization [aOR = 1.1; 95% CI 0.5 to 2.9]. The mean QoL scores was lowest among participants in the cardiopulmonary and depression class [β = -4.8; 95% CI -7.3 to -2.3] followed by the cardiometabolic and arthritis class [β = -3.9; 95% CI -6.4 to -1.4].

## Discussion

In this study, we identified classes of adults with cardiometabolic multimorbidity and assessed the association of different multimorbidity combinations with healthcare utilization and QoL. Our findings show four distinct patterns of multimorbidity: relatively "healthy class" with no multimorbidity: abdominal obesity: cardiometabolic and arthritis class comprising participants with hypertension, type 2 diabetes, stroke, abdominal and general obesity, arthritis and cataract; and cardiopulmonary and depression class including participants with angina, chronic lung disease, asthma, and depression Cardiopulmonary multimorbidity was associated with a higher frequency of outpatient visits and higher odds of hospitalization compared to those with no multimorbidity. However, multimorbidity of cardiometabolic diseases, cataracts and arthritis was associated with a higher frequency of outpatient visits and not hospitalization. Participants with cardiometabolic and cardiopulmonary multimorbidity had poorer quality of life compared to those with no multimorbidity.

**Table 2. Distribution of multimorbidity by sociodemographic characteristics in Ghana.**

| Characteristics [%] | ‡Latent classes of multimorbidity | | | | |
| --- | --- | --- | --- | --- | --- |
| | Class 1: Relatively healthy/no multimorbidity diseases | Class 2: Abdominal obesity | Class 3: Cardiometabolic diseases, arthritis & cataract | Class 4: Cardiopulmonary diseases & depression | P-value |
| N | 1,482 | 1,283 | 166 | 197 | |
| Age [years] | | | | | |
| 50–59 | 50.1 | 57.6 | 31.0 | 35.5 | <0.001 |
| 60–69 | 28.3 | 24.8 | 41.0 | 26.1 | |
| 70+ | 21.7 | 17.6 | 28.0 | 38.4 | |
| Sex | | | | | |
| Male | 76.0 | 18.9 | 22.5 | 41.2 | <0.001 |
| Female | 24.0 | 81.1 | 77.5 | 58.8 | |
| Education | | | | | |
| No formal education | 40.2 | 42.6 | 33.7 | 52.3 | 0.080 |
| Primary | 27.6 | 29.5 | 34.7 | 19.1 | |
| Secondary | 28.6 | 23.9 | 26.5 | 28.1 | |
| Tertiary | 3.7 | 4.1 | 5.1 | 0.6 | |
| Employment | | | | | |
| Public | 8.4 | 7.1 | 6.9 | 9.0 | <0.001 |
| Private | 5.6 | 2.1 | 11.5 | 3.5 | |
| Self-employed | 67.1 | 75.5 | 58.7 | 61.1 | |
| Informal employment | 17.1 | 13.9 | 16.5 | 25.2 | |
| Unemployed | 1.8 | 1.4 | 6.4 | 1.2 | |
| Residence | | | | | |
| Urban | 40.8 | 54.7 | 69.6 | 37.8 | <0.001 |
| Rural | 59.2 | 45.3 | 30.4 | 62.2 | |
| Primary source of care | | | | | |
| Private facility | 7.5 | 9.5 | 21.5 | 10.5 | <0.001 |
| Public facility | 37.9 | 44.2 | 48.5 | 43.8 | |
| Faith-based/charity hospital | 5.0 | 2.8 | 1.8 | 5.4 | |
| [†] Others | 4.2 | 3.7 | 0.2 | 1.1 | |
| Never sought care | 45.4 | 39.8 | 28.0 | 39.2 | |
| Health insurance cover | | | | | |
| Yes | 23.8 | 26.8 | 28.2 | 26.3 | 0.490 |
| No | 76.2 | 73.2 | 71.8 | 73.7 | |

Cells are weighted column percentages.

[†] Other sources of primary care comprise local pharmacies and traditional healers.

IQR; Interquartile range.

‡The multimorbidity clusters included a relatively "healthy class" with no multimorbidity [class 1]: abdominal obesity [class 2]: cardiometabolic and arthritis class comprising participants with hypertension, abdominal and general obesity, arthritis and cataract [class 3]; and cardiopulmonary and depression class including participants with angina, chronic lung disease, asthma, and depression [class 4].

The multimorbidity clusters identified in our study are similar to those in previous studies [49,50]. A systematic review of multimorbidity patterns from 39 studies conducted in 12 countries identified hypertension and arthritis as the most frequent multimorbidity combination [50]. Another study conducted in South Africa found two distinct multimorbidity clusters

**Table 3. Healthcare utilization and quality of life in Ghana [n = 3,128].**

| Characteristics | | Outpatient visits | | | Hospitalized | | Quality of life | | |
|---|---|---|---|---|---|---|---|---|---|
| | | Median | IQR | P-value | Yes | P-value | % | SD | P-value |
| Age [years] | | | | | | | | | |
| | 50–59 | 0 | 1 | <0.001 | 3.6 | 0.028 | 76.3 | 9.5 | <0.001 |
| | 60–69 | 1 | 1 | | 4.6 | | 73.8 | 9.9 | |
| | 70+ | 1 | 1 | | 6.3 | | 68.3 | 12.1 | |
| Sex | | | | | | | | | |
| | Male | 0 | 1 | <0.001 | 1.9 | 0.192 | 75.5 | 10.9 | <0.001 |
| | Female | 1 | 1 | | 2.6 | | 72.4 | 10.3 | |
| Education | | | | | | | | | |
| | No formal education | 0 | 1 | 0.826 | 1.7 | | 71.3 | 10.2 | <0.001 |
| | Primary | 0 | 1 | | 1.4 | | 74.4 | 10.8 | |
| | Secondary | 1 | 1 | | 1.2 | | 76.5 | 10.3 | |
| | Tertiary | 1 | 2 | | 0.2 | | 80.0 | 9.7 | |
| Employment | | | | | | | | | |
| | Public | 1 | 2 | <0.001 | 3.1 | 0.035 | 76.9 | 11.0 | <0.001 |
| | Private | 0 | 1 | | 0.2 | | 75.2 | 10.7 | |
| | Self-employed | 0 | 1 | | 4.4 | | 74.2 | 10.3 | |
| | Informal employment | 1 | 2 | | 6.2 | | 71.3 | 11.5 | |
| | Unemployed | 0 | 2 | | 6.7 | | 70.0 | 11.3 | |
| Residence | | | | | | | | | |
| | Urban | 1 | 2 | <0.001 | 5.3 | 0.068 | 74.9 | 11.1 | <0.001 |
| | Rural | 0 | 1 | | 3.7 | | 73.0 | 10.2 | |
| Primary source of care | | | | | | | | | |
| | Private facility | 1 | 2 | <0.001 | 9.1 | <0.001 | 74.6 | 9.6 | <0.001 |
| | Public facility | 1 | 2 | | 7.6 | | 72.1 | 10.5 | |
| | Faith-based/charity hospital | 1 | 3 | | 10.6 | | 70.6 | 9.9 | |
| | [†] Others | 1 | 1 | | 1.5 | | 70.0 | 12.2 | |
| | Never sought care | | | | | | 76.2 | 10.5 | |
| Health insurance cover | | | | | | | | | |
| | Yes | 1 | 1 | <0.001 | 6.0 | 0.035 | 73.2 | 10.2 | 0.528 |
| | No | 0 | 1 | | 3.9 | | 74.1 | 10.8 | |
| ‡Multimorbidity clusters | | | | | | | | | |
| Class 1: Relatively healthy class/no multimorbidity | | 0 | 1 | <0.001 | 3.8 | 0.128 | 74.7 | 10.4 | <0.001 |
| Class 2: Abdominal obesity | | 1 | 1 | | 4.6 | | 74.5 | 10.2 | |
| Class 3: Cardiometabolic diseases, arthritis & cataract | | 1 | 3 | | 5.2 | | 68.9 | 11.3 | |
| Class 4: Cardiopulmonary diseases & depression | | 1 | 2 | | 8.1 | | 67.5 | 12.4 | |
| Total | | 0 | 1 | | 4.5 | | 73.9 | 10.7 | |

Cells are weighted column percentages.

[†] Other sources of primary care comprise local pharmacies and traditional healers.

IQR; Interquartile range.

‡The multimorbidity clusters included a relatively "healthy class" with no multimorbidity (class 1): abdominal obesity (class 2): cardiometabolic and arthritis class comprised participants with hypertension, abdominal and general obesity, arthritis and cataract (class 3); and cardiopulmonary and depression class included participants with angina, chronic lung disease, asthma, and depression (class 4).

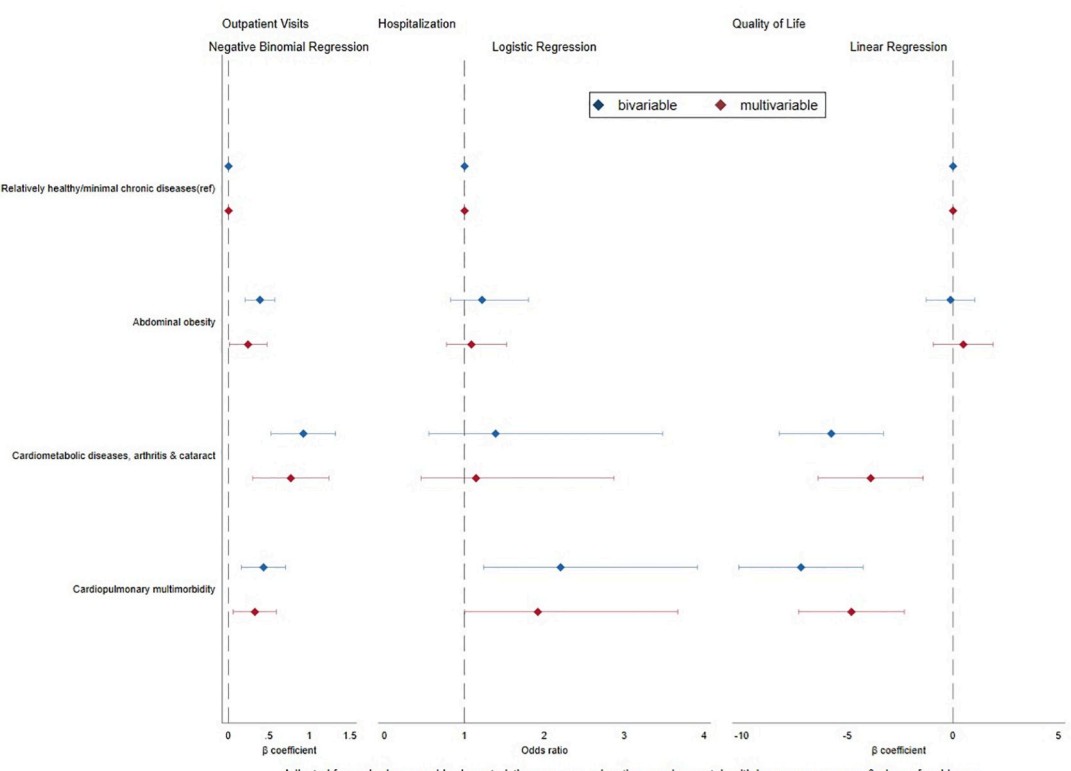

**Fig 3.**

comprising hypertension and diabetes and cardiopulmonary diseases such as angina, asthma and chronic lung disease [49]. In our study: 47.9% of the participants were classified under a relatively "healthy class" with no multimorbidity: 40.7% under the abdominal obesity class: 5.7% under the cardiometabolic and arthritis class and 5.7% under the cardiopulmonary diseases and depression class. The mechanisms that underlie the clustering of cardiopulmonary diseases and depression are not definitive. However, there is strong evidence linking inflammatory markers to both depression and cardiovascular diseases [51], but why these links exist remains unclear.

Systematic reviews conducted by Mullerova et al. [52] and Prados-Torres et al. [53], identified inflammation, stress processes, hypoxia, and environmental risk factors such as air pollution and smoking as the leading risk factors for the clustering of cardiopulmonary diseases such as hypertension, angina, chronic lung disease and asthma [52,53]. Similarly, our previous study on the patterns of cardiometabolic multimorbidity in sub-Saharan Africa identified the clustering of physical inactivity and obesity as one of the leading risk factors for cardiometabolic multimorbidity. However, the study did not include the clustering of cardiometabolic diseases with unrelated conditions such as arthritis, cataract and chronic respiratory diseases [54]. Importantly, the discordant multimorbidity clusters without well-established pathogeneses such as cardiometabolic diseases and arthritis identified in the current study should be studied in the future to elucidate the causal pathways.

Our findings show that the multimorbidity patterns among older adults in Ghana are distinct with important differences with respect to healthcare utilization and QoL. Multimorbidity of cardiometabolic diseases, arthritis and cataract was associated with higher levels of healthcare utilization than cardiopulmonary and depression multimorbidity. However,

cardiopulmonary and depression multimorbidity was associated with the highest odds of hospitalization. Nevertheless, both cardiopulmonary and cardiometabolic multimorbidity were positively associated with poor quality of life compared to participants with no multimorbidity. Although these findings are consistent with previous studies conducted in low and middle-income countries [4,20,55], it is important to note that the existing studies were based on multimorbidity counts without adequate information on specific disease clusters to guide primary care. Unlike the multimorbidity counts, where all morbidities are equally scored irrespective of their relationships, our approach provides crucial insight into the burden of specific multimorbidity clusters that goes beyond counting the number of coexisting chronic conditions. In line with previous studies [4,56–58], there is a possibility that QoL may have deteriorated, partly due to the treatment burden including medication intake, drug management, self-monitoring, lifestyle changes and hospitalization. However, future studies should focus on identifying the underlying causal pathways connecting distinct cardiometabolic multimorbidity clusters, healthcare utilization patterns and QoL.

## Strengths and limitations

This study has three main strengths. First, data are from a nationally representative population-based survey using a standardised WHO-SAGE protocol. Thus, the findings are generalizable to the population of persons aged 50 years and above in Ghana. Second, screening for obesity, hypertension, angina pectoris, arthritis, asthma, chronic lung disease, and depression was based on objective measures comprising direct physical measurement of anthropometrics, BP, symptomatology algorithms and self-reports. Third, the use of LCA and agglomerative hierarchical clustering algorithms in the identification of distinct cardiometabolic multimorbidity clusters provides crucial insights into the patterns of non-random co-occurrence of multimorbidity that goes beyond simple counts used in the majority of previous studies.

The current study has some limitations. First, the screening questions particularly for diabetes, stroke and cataract were based on self-reported history of diagnosis. This may have resulted in the underestimation of the true prevalence of chronic diseases. Second, the current study assessed the association of different cardiometabolic multimorbidity combinations with the frequency of outpatient visits and hospitalization in Ghana. However, the nature of outpatient visits or hospitalization such as routine care or emergencies was not explored. Furthermore, the association of multimorbidity clusters with the cost of care were not investigated. Thus future studies on the economic burden of different cardiometabolic multimorbidity combinations are needed. Third, the number of chronic diseases in the LCA was limited to those included in the SAGE survey in Ghana. This may have excluded other common chronic conditions among older persons, such as dementia, cancers chronic kidney disease, resulting in an underestimation of the multimorbidity burden. Future studies need to include more chronic diseases to increase the external validity. Fourth, the cross-sectional design of the data used in this analysis implies a lack of conclusions regarding the temporality or causation between the multimorbidity classes, healthcare utilization patterns and QoL. Further studies based on longitudinal analysis need to estimate the incidence of transitions between latent classes of cardiometabolic multimorbidity and their impact on healthcare utilization patterns and QoL. Finally, The WHO SAGE data used in this analysis were collected in 2015, and rapid changes in health and socioeconomic circumstances in Ghana are likely to have affected the burden of chronic diseases and quality of life in the last 8 years. Nevertheless, our findings are based on the most recent data we could access and act as a baseline with which to compare future studies on the burden of multimorbidity on healthcare utilization and quality of life in Ghana.

This study has two key policy implications. First, we identified distinct multimorbidity combinations comprising cardiometabolic diseases, arthritis and cataract class and cardiopulmonary and depression class. This may inform the design of multimorbidity treatment guidelines and primary care interventions for cardiometabolic diseases. Given that most of the existing guidelines for the management of chronic diseases in Ghana are single-disease-focused [13], there is a need for a policy discourse on integrated care of discordant cardiometabolic multimorbidity to enable patients to benefit from minimally disruptive care. Second, these results are useful for identifying target populations of people living with cardiometabolic diseases at high risk of outpatient visits, hospitalizations and poor QoL. This is important for planning service delivery capacity, optimization of resources and health system response.

## Conclusions

Our results provide insight into the cardiometabolic multimorbidity clusters and the associated patterns of healthcare utilization and QoL. The findings of this study show that cardiometabolic multimorbidity among older persons in Ghana cluster together in distinct patterns that differ in healthcare utilization and QoL. This evidence may be used in healthcare planning and development of appropriate clinical guidelines for the management of cardiometabolic multimorbidity. Our findings form the basis for, future research on the aetiology and pathogenesis of discordant multimorbidity clusters, and improved policies to address healthcare access and QoL for older persons living with cardiometabolic multimorbidity in sub-Saharan Africa.

## Supporting information

**S1 Table. Symptomatology algorithms.**
(PDF)

**S2 Table. Comparison between latent class models.**
(PDF)

## Author Contributions

**Conceptualization:** Peter Otieno.

**Data curation:** Peter Otieno.

**Formal analysis:** Peter Otieno, Welcome Wami.

**Supervision:** Gershim Asiki, Charles Agyemang.

**Writing – original draft:** Peter Otieno, Gershim Asiki, Calistus Wilunda, Welcome Wami, Charles Agyemang.

**Writing – review & editing:** Peter Otieno, Gershim Asiki, Calistus Wilunda, Welcome Wami, Charles Agyemang.

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
