## [Decision Letter · Decision Letter 0]

27 Mar 2023

PGPH-D-23-00332

Cardiometabolic multimorbidity and associated patterns of healthcare utilization and quality of life: results from the Study on Global AGEing and Adult Health (SAGE) Wave 2 in Ghana.

Dear Dr. Otieno,

Thank you for submitting your manuscript to PLOS Global Public Health. After careful consideration, we feel that it has merit but does not fully meet PLOS Global Public Health’s publication criteria as it currently stands. Therefore, we invite you to submit a revised version of the manuscript that addresses the points raised during the review process.

We look forward to receiving your revised manuscript.

Kind regards,

Nasheeta Peer

Academic Editor

Journal Requirements:

1. Please ensure that Funding Information and Financial Disclosure Statement are matched.

2. In the Funding Information you indicated that no funding was received. Please revise the Funding Information field to reflect funding received.

3. Your manuscript is missing the following sections: Introduction. Please ensure these are present, and in the correct order, and that any references to subheadings in your main text are correct. An outline of the required sections can be consulted in our submission guidelines here:

https://journals.plos.org/globalpublichealth/s/submission-guidelines#loc-parts-of-a-submission

Additional Editor Comments (if provided):

On what basis were the non-cardiometabolic chronic conditions selected for inclusion in this study?

Why were cataracts included?

Lines 124-126: Please check if this sentence is correct. Why were participants deemed to have a condition if they were ‘screened negative’?

Please present Table 1 together with the overall data by men and women as well together with p-values for gender differences.

How do the faith-based/charity organisations differ from public healthcare facilities in terms of fees charged, etc.?

Table 2: Please define the multimorbidity clusters in the table or footnotes - what does minimal multimorbidity refer to? Tables need to be standalone without referring to the details in the main text.

Reviewers' comments:

Reviewer's Responses to Questions

**Comments to the Author**

1. Does this manuscript meet PLOS Global Public Health’s publication criteria? Is the manuscript technically sound, and do the data support the conclusions? The manuscript must describe methodologically and ethically rigorous research with conclusions that are appropriately drawn based on the data presented.

Reviewer #1: Yes

Reviewer #2: Yes

Reviewer #3: Partly

Reviewer #4: Yes

2. Has the statistical analysis been performed appropriately and rigorously?

Reviewer #1: I don't know

Reviewer #2: Yes

Reviewer #3: Yes

Reviewer #4: Yes

3. Have the authors made all data underlying the findings in their manuscript fully available (please refer to the Data Availability Statement at the start of the manuscript PDF file)?

Reviewer #1: Yes

Reviewer #2: Yes

Reviewer #3: No

Reviewer #4: No

4. Is the manuscript presented in an intelligible fashion and written in standard English?

Reviewer #1: Yes

Reviewer #2: Yes

Reviewer #3: Yes

Reviewer #4: Yes

5. Review Comments to the Author

Reviewer #1: The article classified multimorbidity in older adults and assessed the association between these classes with healthcare utilization and quality of life. The results can affect planning of healthcare policies and interventions which will improve healthcare outcomes among older adults.

Even though the manuscript appears sound, the authors should clarify if the following terms refer to the same class to avoid confusion as they were used interchangeable. They should also clearly define these terms in the explanatory variables.

Minimal multimorbidity ( line 34, table 2, line 199, 214, 223,232, 324)

Minimal cardiometabolic multimorbidity (line 179, 211, 216)

The authors stated that one of their strength was screening for chronic diseases which was based on objective measures comprising direct physical measurement of BP, symptomatology algorithms and self -reports (line 249) However they also stated that one of the limitations of the study was screening for chronic diseases which were partially based on self- report.( 254). This looks contradictory

In the conclusion, the sentence on line 276 – 277 should be rephrased to make the message clear.

Reviewer #2: Thank you for the opportunity to review this compelling manuscript. This work represents a meaningful contribution in an understudied area using innovative statistical techniques to identify multimorbidity clusters of potential relevance to health services delivery in a global population.

There are a few areas of major improvements to this manuscript.

- Firstly, the data used in this study are from 2015. While SAGE Wave 3 data from 2018-2019 are still pending, it may potentially be valuable to wait to provide an update on this cluster analysis with newer data, given that the data represented are 8 years old with the prospect of a newer data set. The newer data may provide a more meaningful representation of the current status of the population’s multimorbidity and needs.

- Additionally, it is to be expected that patients with a higher disease burden would receive more outpatient and inpatient services. It would strengthen this analysis to include data on what kind of outpatient visits and inpatient hospitalizations patients who group into clusters received. For example, were there a greater degree of urgent outpatient visits recorded for non-routine reasons such as medication refills, but rather for exacerbations of pathologies? This would strengthen an understanding of what kinds of services are most needed by this population.

- The authors also posit that the clustering of cardiopulmonary diseases may be explained by sociodemographic factors such as smoking, however, data on participants’ smoking history, access to food support, etc. are not provided. Inclusion of this data would strengthen the causal connection, although it may be beyond the scope of what was collected in this study. Regardless, mentioning these variables in the analysis and discussion of why or why not it was included would be valuable.

- Additionally, hypertension is included in class 2 (hypertension and arthritis) and 3 (hypertension, angina, chronic lung disease, and asthma) of the cluster analyses. Given the dual appearance of hypertension within these clusters, it would be important to show any potential interaction between these clusters and whether there is any overlap or competing effects from cluster type and visits/hospitalizations.

Minor revisions include:

- Proofreading needs

- Authors mention the phenomenon of discordant multimorbidity but do not discuss the role of depression in relation to the multimorbid conditions mentioned. Additionally, the authors mention integrated care interventions that would benefit patients with multi morbid conditions, but the paper would be strengthened by inclusion of examples of these interventions.

Given this, I suggest a major revision with opportunity to resubmit to address the above points. Thank you again for the opportunity to review this work.

Reviewer #3: Thank you for the opportunity to review this interesting manuscript. The authors sought to identify classes of cardiometabolic multimorbidity groups among Ghanaian adults aged over 50 years and assess the association of the different multimorbidity combinations with healthcare utilization and quality of life(QoL) using data from the WHO SAGE wave 2. The study adds to the literature on multimorbidity in sub-Saharan Africa and has significant implications on health system planning and policies on integrated guidelines for management of patients with multimorbidity in Ghana. However, I have some comments.

There are several methodological issues.

1. The authors state “Detailed descriptions of sampling methods and data collection procedures have been previously published (26)”, “Data used in the current study were collected using interviewer-administered structured questionnaires (26)”. “Detailed information on the study tools has been published (28).”

The data set and study design used for the study are not available at the cited references “26. World Health Organization. STEPS Manual, STEPS Instrument. Geneva: WHO; 2011, .” and “28. Kowal P, Chatterji S, Naidoo N, Biritwum R, Fan W, Lopez Ridaura R, et al. Data resource profile: the World Health Organization Study on global AGEing and adult health (SAGE). International journal of epidemiology. 2012;41(6):1639-49.” respectively.

The WHO STEPS Manual does not provide the data set for the present study. Also the publication by Kowal P et al. seems to discuss WHO SAGE wave 0 and 1.

The authors can provide appropriate references for the stated methodology or outline them in detail in the current manuscript.

2. The data availability statement also states that data is publicly available on the microdata repository of the WHO (https://apps.who.int/healthinfo/systems/surveydata/index.php/catalog). However, the data on this repository does not include the data used for the present study. If the data is available on specific request that should rather be made known by the authors.

3. Concerning the sample size and sampling:

I am curious to know what informed the authors’ decision to exclude 2,037 of the 3,575 original study sample because they had not used outpatient care in the 12 months preceding the survey; and why 12 months in particular? My thought is that, “not having used outpatient care” is a pattern of healthcare utilization, and this idea has completely been jettisoned.

Frequency of hospitalization is independent of frequency of outpatient visit by their definition, so if persons with no outpatient visit in the preceding 12 months are excluded, essentially persons who may have been hospitalized in that period but had no outpatient visit are also excluded. If hospitalization is implied to be only after outpatient visit, then frequency of hospitalizations will be a subset of frequency of outpatient attendance. This seems to be a major flaw in sampling.

4. On the measurement and definition of variables:

The frequency of hospitalization as defined is expected to have been reported as a continuous variable. However, in Table 2, it is categorized as “greater than or equal to 3” and by inference“1 to 2” outpatient visits. What informs this categorization at cut-off of 3? And the authors must at least state this categorization in the methodology.

In the report of the results, line 186, it is interpreted as participants who visited outpatient clinic “more than three times” which is inaccurate. It should rather be stated as “three or more times”.

The frequency of hospitalization as defined is expected to have been reported as a continuous variable. However, in Table 2, it is categorized as “Yes” and by inference “No”.

In the report of the results, line 186, it is interpreted as participants who were hospitalized at least once. This is not consistent with their methodology.

The WHOQoL instrument gives results as a continuous variable(percentage between 0-100%). Can the authors explain why they categorized the results of QoL? …and in tertiles? Were the results normally distributed? Also why not as a binomial into poor and good quality of life because essentially the report lumped moderate and good quality of life as against poor quality of life in the results, lines 202-205 and discussion, lines 215-217. Can the result of QoL be reported or represented as odds of poor quality of life then.

History of medical conditions as well as screening questions for the chronic diseases were based on self-report which has been acknowledged as a major limitations of this study. Since hypertension was objectively screened for, the prevalence appears representative. Knowing that “hypertension and diabetes account for the highest burden of multimorbidity in Ghana (5) ” the prevalence of diabetes is expected to also be high, however, conditions that were not screened for like diabetes, stroke and cataract are likely to be under-reported.

5. In the analysis:

The authors did not state the level of p-values at which analysed variables were considered statistically significant.

Other minor issues:

1. The authors did not comply with using square brackets [] for citations.

2. In Table 2, † did not appear in the legend.

3. Under findings of latent class analysis:

What really is the meaning and import of “minimal cardiometabolic multimorbidity” in the present study? This should be addressed in the discussion as well. Were there any participants with no single morbidity or multimorbidity?

4. In the discussion:

Line 235 to 236_ the study by Sum G et al.(22) as the authors cite, did not look at number of comorbidity only but also elucidated clusters of multimorbidities which (hypertension + arthritis) seemed to a predominant multimorbidity.

Line 239 to 241_ can the authors explain or give some examples of the several underlying mechanisms that could explain the associations in the present study?

5. In the references:

15. …Journal of Morbidity and Comorbidity not “Journal of Multimorbidity and Multimorbidity.”

Reviewer #4: Overall, this paper is well-written. This study identified three groups of patients with varying probabilities on nine diseases using a latent class analysis. The researchers also examined the associations between the class memberships and three outcome variables (outpatient visits, hospitalization, and QoL). The topic is important, and the results are interesting. However, there are some major comments or suggestions regarding the methods and results that could be considered to improve the quality of this paper.

Major:

1. Partial list of cardiometabolic diseases or conditions was included in this study. Other important diseases or conditions such as obesity, dyslipidemia, and chronic kidney disease were not included. If data on some or all of these diseases are available, it would be helpful to add them to the analysis. If not, it is suggested to discuss this as a limitation.

2. Because LCA is the main analysis in this study to identify latent classes of participants with multiple cardiometabolic diseases and non- cardiometabolic diseases, more details of results from the LCA models could be provided to help readers to understand the relationships between the sociodemographic and health characteristics and latent classes. For example, how were the nine disease variables coded and modeled in LCA? What is the distribution of the sociodemographic health characteristics of the study participants (as listed in Table 1) in the three latent classes?

3. What is the meaning of the beta coefficients from the negative binomial regression (Line 200-201)? What is the clinical significance of these results? This information is important for readers to interpret the findings.

4. How was QoL coded and analyzed in ordinal logistic regression? For example, was QoL coded as low = 0, moderate=1, and high=2 or vice versa in the ordinal logistic regression model? It is important to indicate this information in order to understand the direction and magnitude of the associations.

Minor:

1. More details on the LCA modeling process could be added to the Method section (Line 134-142) to better assist readers in understanding how the latent classes were determined and how the participants were classified into each of the three latent classes. For example, the two sentences on Line 141-142 might be difficult to be understood by readers. What are the posterior probabilities? How are they related to the determination of class memberships?

2. Reference # 26 indicates WHO STEP Manual 2011 (Line 380). However, data from the WHO Study on Global AGEing and Adult Health (SAGE) Wave 2 were used (Line 83).

6. PLOS authors have the option to publish the peer review history of their article (what does this mean?). If published, this will include your full peer review and any attached files.

**Do you want your identity to be public for this peer review?** For information about this choice, including consent withdrawal, please see our Privacy Policy.

Reviewer #1: No

Reviewer #2: No

Reviewer #3: **Yes: **Dr. Kwadwo Faka Gyan

Komfo Anokye Teaching Hospital

Kumasi, Ghana

Reviewer #4: No

---

## [Decision Letter · Decision Letter 1]

8 Jun 2023

PGPH-D-23-00332R1

Cardiometabolic multimorbidity and associated patterns of healthcare utilization and quality of life: results from the Study on Global AGEing and Adult Health (SAGE) Wave 2 in Ghana.

Dear Dr. Otieno,

Thank you for submitting your manuscript to PLOS Global Public Health. After careful consideration, we feel that it has merit but does not fully meet PLOS Global Public Health’s publication criteria as it currently stands. Therefore, we invite you to submit a revised version of the manuscript that addresses the points raised during the review process.

We look forward to receiving your revised manuscript.

Kind regards,

Nasheeta Peer

Academic Editor

Journal Requirements:

a. State what role the funders took in the study. If the funders had no role in your study, please state: “The funders had no role in study design, data collection and analysis, decision to publish, or preparation of the manuscript.”

b. If any authors received a salary from any of your funders, please state which authors and which funders.

Additional Editor Comments (if provided):

Lines 199-200: Was the higher prevalence of chronic lung diseases in women vs. men expected? What was this due to? Generally, in Africa, it’s higher in men because of their higher rates of smoking.

I suggest that you rephrase this sentence and only report the conditions that are significantly different between the sexes.

Line 207: Please clarify what is meant by “minimal chronic diseases’ – is it no chronic diseases?

Line 208: Does class two include abdominal obesity alone, with no other comorbidities?

Please also clarify in table on Page 11. This should be Table 2 and not 3 – line 233. Please also correct table numbering on line 236 and 245.

Table 1: I suggest presenting the column with the total (both sexes) data first followed by the male/female columns because the p-value relates to the latter columns. This will read better.

P values in all tables should be to 3 decimal places, please.

Line 298: Please add 'diseases' after 'cardiometabolic'.

Line 322: Please add: ‘particularly for diabetes’ after ‘diseases'.

Reviewers' comments:

Reviewer's Responses to Questions

**Comments to the Author**

1. If the authors have adequately addressed your comments raised in a previous round of review and you feel that this manuscript is now acceptable for publication, you may indicate that here to bypass the “Comments to the Author” section, enter your conflict of interest statement in the “Confidential to Editor” section, and submit your "Accept" recommendation.

Reviewer #1: All comments have been addressed

Reviewer #3: All comments have been addressed

Reviewer #4: All comments have been addressed

2. Does this manuscript meet PLOS Global Public Health’s publication criteria? Is the manuscript technically sound, and do the data support the conclusions? The manuscript must describe methodologically and ethically rigorous research with conclusions that are appropriately drawn based on the data presented.

Reviewer #1: Yes

Reviewer #3: Yes

Reviewer #4: Yes

3. Has the statistical analysis been performed appropriately and rigorously?

Reviewer #1: I don't know

Reviewer #3: Yes

Reviewer #4: Yes

4. Have the authors made all data underlying the findings in their manuscript fully available (please refer to the Data Availability Statement at the start of the manuscript PDF file)?

Reviewer #1: Yes

Reviewer #3: Yes

Reviewer #4: Yes

5. Is the manuscript presented in an intelligible fashion and written in standard English?

Reviewer #1: Yes

Reviewer #3: Yes

Reviewer #4: Yes

6. Review Comments to the Author

Reviewer #1: The authors have addressed all the comments raised.

Reviewer #3: (No Response)

Reviewer #4: Thank you for addressing the comments and suggestions raised on the previous version. The major revisions are appropriate and adequate. Adding the abdominal obesity variable to the LCA modeling is helpful. The results from the 4-class model are more meaningful than the previous results from the 3-class model. The results on the associations between latent class memberships and the healthcare utilization patterns and QoL have more clinical significance in this 4-class model compared to the previous 3-class model. The following are some additional comments and suggestions for you to consider.

Major:

1. The results of the supplementary hierarchical cluster analysis seem to be unrelated to the objectives of the study and did not provide much support to the results of the LCA modeling either. They might cause unnecessary confusions to the readers. Suggest removing this method and the related results from the paper.

2. The labeling for latent class 3 and 4 seems not to reflect the highest probabilities of diseases within each latent class sufficiently. For example, in class 3, hypertension and abdominal obesity have the highest probabilities, followed by general obesity and arthritis. In class 4, asthma and chronic lung disease have the highest probabilities, followed by hypertension, abdominal obesity, and angina. As you show in Table 1, abdominal obesity has the highest prevalence, particularly in females. Hypertension has the second highest prevalence. Therefore, abdominal obesity and hypertension could be the two major diseases that could potentially be related to other comorbidities in this population.

3. I would suggest using the following method to simplify the labeling of the four latent classes based on the major diseases with the highest probabilities within each latent class: class 1 - no comorbidity; class 2 - abdominal obesity; class 3 - cardiometabolic comorbidities; class 4 - cardiopulmonary comorbidities.

7. PLOS authors have the option to publish the peer review history of their article (what does this mean?). If published, this will include your full peer review and any attached files.

**Do you want your identity to be public for this peer review?** For information about this choice, including consent withdrawal, please see our Privacy Policy.

Reviewer #1: No

Reviewer #3: **Yes: **Dr. Kwadwo Faka Gyan

Komfo Anokye Teaching Hospital

Kumasi, Ghana

Reviewer #4: No

---

## [Decision Letter · Decision Letter 2]

10 Jul 2023

Cardiometabolic multimorbidity and associated patterns of healthcare utilization and quality of life: results from the Study on Global AGEing and Adult Health (SAGE) Wave 2 in Ghana.

PGPH-D-23-00332R2

Dear Dr. Otieno,

We are pleased to inform you that your manuscript 'Cardiometabolic multimorbidity and associated patterns of healthcare utilization and quality of life: results from the Study on Global AGEing and Adult Health (SAGE) Wave 2 in Ghana.' has been provisionally accepted for publication in PLOS Global Public Health.

Best regards,

Nasheeta Peer

Academic Editor

Lines 34-38: This is unclear; while the details have been provided in the main text, it also needs to be clear in the Abstract. Please amend the 2nd class, if correct, as “abdominal obesity only (40.7%)”; the 3rd class as “any combination of…” and for class 3 and 4, replace “and” with “or” where appropriate.

Please also make the corresponding amendments in the main text.

Reviewer Comments (if any, and for reference):

Reviewer's Responses to Questions

**Comments to the Author**

1. If the authors have adequately addressed your comments raised in a previous round of review and you feel that this manuscript is now acceptable for publication, you may indicate that here to bypass the “Comments to the Author” section, enter your conflict of interest statement in the “Confidential to Editor” section, and submit your "Accept" recommendation.

Reviewer #2: All comments have been addressed

Reviewer #4: All comments have been addressed

2. Does this manuscript meet PLOS Global Public Health’s publication criteria? Is the manuscript technically sound, and do the data support the conclusions? The manuscript must describe methodologically and ethically rigorous research with conclusions that are appropriately drawn based on the data presented.

Reviewer #2: Yes

Reviewer #4: Yes

3. Has the statistical analysis been performed appropriately and rigorously?

Reviewer #2: Yes

Reviewer #4: Yes

4. Have the authors made all data underlying the findings in their manuscript fully available (please refer to the Data Availability Statement at the start of the manuscript PDF file)?

Reviewer #2: Yes

Reviewer #4: Yes

5. Is the manuscript presented in an intelligible fashion and written in standard English?

Reviewer #2: Yes

Reviewer #4: Yes

6. Review Comments to the Author

Reviewer #2: Thank you for submitting this updated manuscript. This version is much improved and significantly addresses this reviewer's prior concerns regarding the suitability of use of the SAGE Wave 2 data, the potential etiologies of connection between the disease states in the multimorbidity clusters, as well as need for proofreading.

There remain areas of remaining proofreading for consistency as well eliminating redundant language to reduce the overall length of the submission, however, I am pleased to recommend that this manuscript be accepted for its meaningful contributions to improving service planning and delivery.

Reviewer #4: Thank you for addressing the comments and suggestions. The revisions are appropriate and adequate.

7. PLOS authors have the option to publish the peer review history of their article (what does this mean?). If published, this will include your full peer review and any attached files.

**Do you want your identity to be public for this peer review?** For information about this choice, including consent withdrawal, please see our Privacy Policy.

Reviewer #2: **Yes: **Ramya Sampath, MD

Reviewer #4: No
